# Physical activity and handgrip strength in patients with mild, moderate and severe haemophilia: Impacts on bone quality and lean mass

Pia Ransmann[1], Marius Brühl[2], Jamil Hmida[2], Georg Goldmann[3], Johannes Oldenburg[3], Frank Alexander Schildberg[2], Robert Ossendorff[2], Fabian Tomschi[1], Alexander Schmidt[1], Thomas Hilberg[1], Andreas Christian Strauss[2]*

1 Department of Sports Medicine, University of Wuppertal, Wuppertal, Germany, 2 Department of Orthopedics and Trauma Surgery, University Hospital Bonn, Bonn, Germany, 3 Institute for Experimental Haematology and Transfusion Medicine, University Hospital Bonn, Bonn, Germany

* andreas.strauss@ukbonn.de

## Abstract

### Background

Patients with haemophilia (PwH) might be restricted in physical activity (PA) depending on the severity phenotype. It is well-known that PA affects overall health including bone quality. This study aims to evaluate the level of PA within the different haemophilia severity phenotypes and to elaborate on the interplay of PA in regard to bone quality (bone mineral density (BMD) and trabecular bone score (TBS)) as well as lean mass.

### Methods

This investigation was part of a large prospective single-center cohort study examining the relation between haemophilia and osteoporosis registered at clinicaltrials.gov (ID: NCT04524481). PwH underwent a dual x-ray screening using Horizon™ to examine BMD, TBS, and lean mass. Step activity was tracked electronically for seven consecutive days after clinical examination, supported by a self-reported activity diary for seven days. Handgrip strength was examined as an overall fitness proxy.

### Results

Data of 223 patients with either mild (N=45), moderate (N=46), or severe (N=132) haemophilia A or B, aged 43.6±15.6 years were analyzed. There was no significant difference in objective (p=0.162) and subjective (p=0.459) PA levels between severity phenotypes. The most frequent type of PA in all severities was walking (n=72, 53.3%) and cycling (n=60, 44.4%). Step activity positively correlated with TBS (rho=0.202, p=0.005) and lean mass positively correlated with BMD (rho=0.309, p<0.001). Handgrip strength correlated with BMD (rho=0.361, p<0.001) as well as TBS (rho=0.221, p=0.021) and lean mass (rho=0.287, p=0.003).

**Data availability statement:** All relevant data are within the manuscript and its Supporting Information files.

**Funding:** Bayer Vital GmbH. The funders had no role in study design, data collection and analysis, decision to publish, or preparation of the manuscript.

**Competing interests:** Pia Ransmann received speakers' fees and travel support from Takeda Pharmaceuticals as well as travel support from Swedish Orphan Biovitrum GmbH. Marius Brühl received travel support from Takeda Pharmaceuticals as well as travel support from Swedish Orphan Biovitrum GmbH. Jamil Hmida has no conflict of interest to declare. Georg Goldmann has received advisory board and travel expenses from Bayer, BioMarin, Biotest, Chugai Pharmaceutical Co, Ltd, CSL Behring, Grifols, LFB, Novo Nordisk, Octapharma, Pfizer, F. Hoffmann-La Roche Ltd, Swedisch Orphan Biovitrum, and Takeda. Johannes Oldenburg has received research funding from Bayer, Biotest, CSL Behring, Octapharma, Pfizer, Swedish Orphan Biovitrum, and Takeda and has received consultancy, speakers bureau, honoraria, scientific advisory board and travel expenses from Bayer, Biogen Idec, Biomarin, Biotest, CSL Berhing, Chugai, Freeline, Grifols, LFB, Novo Nordisk, Octapharma, Pfizer, Roche, Sanofi, Spark Therapeutics, Swedish Orphan Biovitrum, and Takeda. Frank Schildberg has no conflict of interest to declare. Robert Ossendorff has no conflict of interest to declare. Fabian Tomschi has received speakers' fees and travel support from Takeda and received an educational grant from Swedish Orphan Biovitrum. Alexander Schmidt received speakers' fees and travel support from Takeda Pharmaceuticals as well as travel support from Swedish Orphan Biovitrum GmbH. Thomas Hilberg has received research funding from Biotest, Intersero, Swedish Orphan Biovitrum, Roche and Novo Nordisk and has received consultancy, speakers bureau, honoraria, scientific advisory board and travel expenses from Bayer, Biotest, Chugai, Novo Nordisk, Pfizer, Sanofi, Swedish Orphan Biovitrum and Takeda. Andreas Strauss has received research funding from Bayer, Swedish Orphan Biovitrum and Takeda and has received consultancy, speakers bureau, honoraria, scientific advisory board and travel expenses from Bayer, Biotest, CSL Behring, Novo Nordisk, Swedish Orphan Biovitrum and Takeda.

## Conclusion

PA does not differ significantly between the severity phenotypes. The majority of PwH in all severity phenotypes performed low-impact PA, which is most likely insufficient to positively affect BMD. However, handgrip strength correlates with BMD and TBS. Despite restrictions in movement function or reduced BMD, it is of major importance to promote PA to maintain a healthy or even increase bone quality.

## Introduction

Haemophilia is a hereditary bleeding disorder, caused by a deficiency of factor VIII (haemophilia A) or IX (haemophilia B) [1,2]. Depending on the factor activity, patients with haemophilia (PwH) can suffer from either mild (5–15%), moderate (1–5%) or severe (<1%) haemophilia, which commonly determines the bleeding rates [3]. Joint bleeding is a primary consequence of haemophilia, often leading to haemophilic arthropathy, characterized by pain and restricted range of motion [4].

Nevertheless, it is well-known that physical activity (PA) represents a key factor for a healthy lifestyle with various beneficial effects on organs, muscles, and bones [5]. Though, a physically active lifestyle was not recommended for PwH until the 1970 because of a suspected increased bleeding risk at the time; PA in PwH is nowadays indispensable [6,7]. This is due to better treatment options but also to the risen awareness of general beneficial effects of exercising on the patient's health [8]. Even though PA is highly promoted in the meantime, scientific data on the extent of activity levels in haemophilic populations is controversial and data investigating the differentiation of PA levels between severity phenotypes is limited [9–12]. However, it has been described that patients affected by arthropathy show lower PA levels especially regarding higher intensities. Additionally, many PwH are affected by kinesiophobia, which is considered as the fear of pain and/or injury due to movement, especially in regard to high-impact sports because of the increased bleeding risk [13,14]. Of course, factor activity plays a key role in managing the risk profile associated with high-impact activities, though recent research suggests that PA-induced bleeding rates are relatively low [15,16].

However, in addition to favorable effects on the general individual health, PA is of major importance to maintain or gain muscle mass, also referred to as lean mass [17,18]. Lean mass is positively associated with reduced cardiometabolic risk factors, improved self-determinate aging, and healthier bone metabolism [19,20]. The overall muscle strength can be easily evaluated by assessing handgrip strength. In patient populations, such as PwH where joint pain and arthropathy can limit movement, measuring handgrip strength offers a practical way to assess overall strength without requiring more strenuous testing that might exacerbate joint issues [21,22]. Further, It has been shown that handgrip strength correlates with bone quality and it has been shown to be a valid predictor of fall risk [23]. Risk of falling is an essential concern for patients with reduced bone quality and reduced lean mass. Bone quality is represented by bone mineral density (BMD), which is positively influenced by PA [24], and trabecular bone score (TBS), an innovative measure that evaluates the trabecular microarchitecture of the bone. Good trabecular microarchitecture reduces fracture risk, despite low BMD [25]. Current knowledge on TBS in PwH is very scarce with only minimal evidence currently available [26,27]. Previous research of our group indicates that lean mass and BMD are notably reduced in patients with severe haemophilia and that TBS appears to be predominantly normal across different severity phenotypes [28]. It is known that TBS is reduced by ageing and increased weight, but further potential influencing lifestyle factors on TBS are not well-studied [29].

However, the complex interrelationships between PA and handgrip strength, TBS, lean mass, and BMD within a large haemophilic cohort of all three severity phenotypes have not been investigated yet. Based on these considerations, this study aims to fill this crucial research gap and the following three research questions were formulated:

(1)  How does the severity phenotype influence the level of PA in PwH?

(2)  How does PA correlate with TBS, BMD, and lean mass in PwH?

(3)  How do lean mass and handgrip strength impact BMD and TBS in PwH?

## Methods

### Study design and participants

This investigation was part of a large prospective osteoporosis and haemophilia study. Data regarding BMD and lean mass of the entire sample (n = 255) was previously published [28,30]. In the current publication, these parameters are presented for the sample of the present study (n = 223). Adult male patients with either mild, moderate or severe haemophilia A or B were included. Patients with other bleeding disorders or younger than 18 years were excluded from this investigation. In agreement with the haemophilia joint health score (HJHS) manual, patients who experienced joint bleedings in the past two weeks were also excluded [31]. This study was conducted in accordance with the principles of good clinical and ethical practice and was acceded by the local ethic committee (339/19). Ethical approval for this study was gained at 14th of November 2019, though because of the Covid-19 pandemic, recruitment process started at 19th of August 2020 and ended at 29th of September 2022 at the University Hospital Bonn, Germany (registered at clinicaltrials.gov (ID: NCT04524481)). Along with the Declaration of Helsinki, all participants gave written informed consent after being informed about the study protocol.

### Data acquisition

To evaluate a patient's daily PA level, PwH were given an electronic activity tracker (Fitbit Alta Hr, Fitbit Inc., San Francisco, USA), which was worn at the wrist for seven consecutive days after the clinical examination. The activity tracking for seven days is in line with the recommendations to collect accelerometry data over multiple days to achieve a reliable estimate of an individual's habitual PA [32–34]. The average number of steps taken within one day over the seven-day observation period (objective PA) was used for further analysis. In parallel, subjects kept an activity diary on the same seven days. Here, PwH were instructed to report performed type of PA as well as the respective duration in minutes (subjective PA) to assess daily time spend on PA. Daily activities such as housekeeping or shopping were not considered as PA and therefore excluded from further analysis. Moreover, handgrip strength, used as a correlate for the overall fitness level, was measured in n = 102 PwH using a hand dynamometer (Baseline, White Plains, NY) [35,36]. The assessment was performed bilaterally, three times in a sitting position with 90° elbow flexion. The mean of the left and right hand was used for further analysis.

The clinical examination entailed two major procedures: First, a dual x-ray (DXA) screening using Horizon™ (Hologic, USA) of the whole body, the left hip and lumbar spine was conducted. The whole-body screening revealed the subject's lean mass (g) and the left hip (neck) was used to determine the BMD (g/cm²). Additionally, the software TBS iNsight® (V. 3.1.2. Medi Maps; Switzerland) revealed the TBS based on the DXA of the lumbar spine. The TBS is measured in score points, which are classified as "normal" (TBS ≥ 1.31), "partially degraded"

(TBS between 1.30 and 1.24), and "degraded" (TBS ≤ 1.23) [37]. Second, the orthopedic joint status was clinically examined via the HJHS (Version 2.1; maximum score 124 points, 20 points × 6 joints, plus 4 points assigned to global gait; higher values indicating worse joint status), which examines the elbows, knees and ankles in regard to swelling, muscle atrophy, crepitation, stability, pain, muscle strength and range of motion [38].

Via a self-established anamnesis questionnaire, anthropometric data as well as data regarding the pharmacological treatment regime were additionally gathered.

## Statistics

Descriptive statistics were calculated based on the severity of PwH and in total. The IBM© Statistical Package for the Social Sciences 29 (Armonk, NY, USA) for Mac was used for all statistical analyses. Tests for normality by Kolmogorov-Smirnov were conducted, revealing no normal distribution, which was confirmed by visual analysis of Q-Q plots. Hence, data are presented as median [1st quartile, 3rd quartile]. Thus, the Kruskal-Wallis-Test was used to examine between-group differences. In case of significant differences, Bonferroni correction was used for alpha-adjustment. To analyze potential influencing factors, Spearman's rho was calculated for correlation analyses. Here, rho = 0.10 equals a weak correlation, rho = 0.30 represents a moderate correlation and rho = 0.50 is considered as a strong correlation [39].

Regarding research questions 2 and 3, subjects were further classified into two groups based on their activity level (steps/day and PA in minutes per day (upper 50 percent, lower 50 percent)) as well as based on the amount of lean mass (upper 50 percent, lower 50 percent). A supplementary analysis was performed to evaluate group differences between patients who perform strength training compared to patients not performing strength trainings. The Mann-Whitney-U-Test was used to statistically analyze these group differences. A significance level of $p \leq 0.05$ (95% confidence interval) was established.

## Results

Overall, 223 PwH were recruited and analyzed in this study. PwH showed a median age of 43 [30, 56] years. Data of patients with mild (n = 45), moderate (n = 46) and severe (n = 132) haemophilia A (n = 193) or B (n = 30) were analyzed (see Table 1). Results of the HJHS differed

**Table 1. Anthropometric and descriptive data of patients with haemophilia.**

| Variables | Severe (n = 132) | Moderate (n = 46) | Mild (n = 45) | Total (n = 223) | p-value |
|---|---|---|---|---|---|
| **Age** (years) Median [Q1, Q3] | 40 [29, 54] | 49 [32, 59] | 45 [29, 57] | 43 [30, 56] | 0.192 |
| **Weight** (kg) Median [Q1, Q3] | 82 [74, 91] | 88 [80, 96] | 84 [78, 95] | 84 [76, 94] | 0.074 |
| **BMI** (kg/m²) Median [Q1, Q3] | 25.2 [23.1, 27.8] | 25.9 [23.6, 28.6] | 26.7 [23.9, 29.1] | 25.6 [23.6, 28.3] | 0.149 |
| **Haemophilia form** (n) | A: B: | A: B: | A: B: | A: 193 B: 30 | |
| **Viral comorbidities** (n) | HIV: 29 HEP C: 23 | HIV: 2 HEP C: 3 | HIV: 3 HEP C: 1 | HIV: 34 HEP C: 27 | n/a |
| **HJHS** (score) Median [Q1, Q3] | 18* [6, 34] | 7* [5, 13] | 7* [4, 12] | 10 [5, 28] | <0.001* |

Explanation: BMI = Body Mass Index; HJHS=Haemophilia Joint Health Score [2.1]; Kruskal-Wallis-tests were conducted; * indicates significant difference, Bonferroni post-hoc analysis revealed significant differences regarding the HJHS: severe-mild: p ≤ 0.001, severe-moderate: p ≤ 0.001.

significantly between the severity phenotypes (p < 0.001). Bonferroni post-hoc testing revealed that patients with severe haemophilia had a significantly worse joint status compared to patients with moderate (p < 0.001) or mild (p < 0.001) haemophilia.

To address research question 1), objective and subjective PA data were evaluated.

A total of n = 203 patients returned the activity diary, of which n = 72 (35.4%) PwH reported to not being physically active at all. 6 (3%) PwH did not indicate the duration of performed PA in minutes. Out of the remaining 125 PwH, median PA in minutes per day was 42 [22, 81]. The activity diary was further evaluated regarding types of performed PA. In total, the most frequent type of PA in all severity phenotypes was walking (n = 72, 53.3%) followed by cycling (n = 60, 44.4%) and strength training (n = 37, 27.4%; see Table 2).

The activity tracker analysis showed a median step activity/day of 7392 [4981,10579] within the whole study cohort, which did not differ between haemophilia severity types (p = 0.162), analogous to the subjective PA, which neither differed between severity phenotypes (p = 0.459; see Fig 1). In detail, patients with mild haemophilia showed a median of 8466 [5219, 10785] steps per day and indicated to be physically active for 42 [30, 85] minutes per day. Patients with moderate haemophilia accumulated a median of 7040 [5530,11472] steps per day and

**Table 2. Type of sports conducted differentiated by haemophilia severities.**

| PA type | | N (%) | Duration (minutes/week) |
|---|---|---|---|
| **Severe (n = 113)** | Walking | 39 (34.5) | 252 [112,415] |
| | Cycling | 32 (28.3) | 135 [63,255] |
| | Strength Training | 19 (16.8) | 205 [82,242] |
| | Bodily exercising | 10 (8.5) | 60 [33,217] |
| | Jogging | 8 (7.1) | 30 [30,86] |
| | Physiotherapy | 7 (6.2) | 60 [60,240] |
| | Swimming | 6 (5.3) | 90 [32,170] |
| | Diverse[1] | 11 (9.7) | n/a |
| | None | 44 (38.9) | n/a |
| **Moderate (n = 43)** | Walking | 20 (46.5) | 210 [120,274] |
| | Cycling | 17 (39.5) | 90 [41,240] |
| | Strength training | 7 (16.3) | 120 [90,295] |
| | Bodily exercising | 4 (9.3) | 170 [130,270] |
| | Diverse[2] | 11 (25.5) | n/a |
| | None | 14 (32.5) | n/a |
| **Mild (n = 41)** | Walking | 13 (31.7) | 175 [102,594] |
| | Strength training | 11 (26.8) | 150 [69,270] |
| | Cycling | 11 (26.8) | 60 [40,180] |
| | Hiking | 7 (17.1) | 180 [120,265] |
| | Bodily exercising | 5 (12.2) | 85 [67,187] |
| | Diverse[3] | 12 (29.3) | n/a |
| | None | 14 (35.1) | n/a |

Explanation: Data presented as absolute numbers and median [Q1, Q3]; PA=physical activity; multiple answers were possible; n/a = not applicable, bodily exercising can include rehabilitation sports as well as functional training.

[1] = Golf (n = 2), Dancing (n = 2), Hiking (n = 2), E-sports (Nintendo Switch; n = 1), Stand up Paddling (n = 1), Nordic Walking (n = 1), Tennis (n = 1), Soccer (n = 1);

[2] = Physiotherapy (n = 3), Hiking (n = 3), Jogging (n = 1), Horseback riding (n = 1), Golf (n = 1), Swimming (n = 1) Nordic Walking (n = 1);

[3] = Jogging (n = 4), Physiotherapy (n = 2), Skating (n = 1), Basketball (n = 1), Badminton (n = 1), Nordic Walking (n = 1), Pilates (n = 1), Gymnastics (n = 1).

stated to be physically active for 42 [13, 83] minutes per day. Patients with severe haemophilia conducted 7095 [4757, 9801] steps per day with a median of 45 [22, 71] minutes per day of self-reported PA.

Concerning handgrip strength, a statistical difference between the severity phenotypes (p = 0.030) was observed but could not be confirmed by Bonferroni post-hoc testing. A significant correlation was further observed between handgrip strength and HJHS (rho = -.362, p < 0.001). The analysis of the influence of HJHS on PA revealed inverse correlation between HJHS and objective PA (rho = -0.239, p < 0.001), though no influence on subjective PA (r = -0.059, p = 0.489). In addition, the comparison of BMD, TBS and lean mass between the severity phenotypes, displayed in Fig 1, indicates that BMD was significantly lower in patients with severe haemophilia compared to patients with mild haemophilia (p = 0.007, post hoc p = 0.006). Also, lean mass was significantly reduced in patients with severe haemophilia compared to patients with moderate haemophilia (p = 0.004, post hoc p = 0.012). However, TBS did not differ within severity phenotypes (p = 0.234).

Supporting information for Fig 1 can be found as supplementary material (S1).Regarding research question 2) and 3), Spearman correlation analyses between objective and subjective PA, handgrip strength, TBS, BMD and lean mass were conducted. The results are displayed

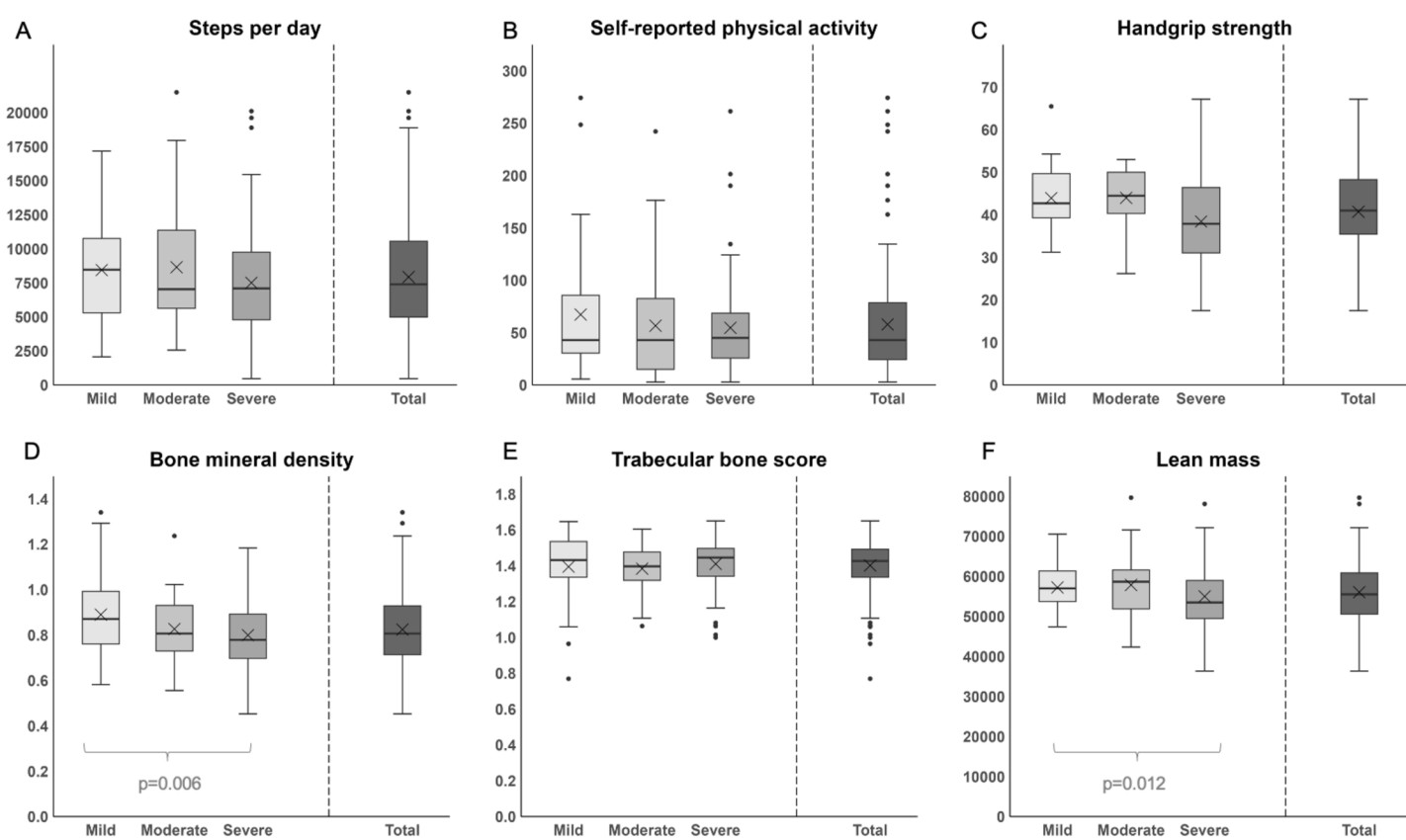

**Fig 1. Boxplots showing the different severity phenotypes regarding physical activity, bone quality, lean mass and handgrip strength.** Explanation: Boxplots of steps per day (in thousands; A; n = 223), self-reported physical activity per day (in minutes; B; n = 125), handgrip strength (in kilograms; C; n = 102)), Bone mineral density (in grams/cm²) of the left neck (D; n = 201)), trabecular bone score (Normal TBS ≥ 1.31, partially degraded between 1.30 and 1.24, degraded TBS ≤ 1.23; E; n = 194), and lean mass (in grams; F; n = 180). The central box signifies the interquartile range (IQR), with the mean displayed as the solid horizontal line and the median as X within the box. Whiskers display 1,5*IQR. Outliers are presented individually as dots.

in Fig 2. Concerning research question 2), a positive correlation was found between objective PA (steps per day) and TBS (Spearman's rho = 0.202, p = 0.005). The Mann-Whitney-U-Test supported these findings, indicating a higher TBS of the patients with a higher step activity level per day (TBS median 1.442 [1.362; 1.527]) compared to the lower step activity level per day (TBS median 1.413 [1.295; 1.485]; p = 0.015). There was no significant difference regarding BMD or lean mass (p > 0.103) neither considering objective nor subjective PA. Further, a positive correlation was observed regarding handgrip strength and objective PA (rho = 0.231, p = 0.020). There was no association between hand grip strength and subjective PA (rho = 0.101, p = 0.453).

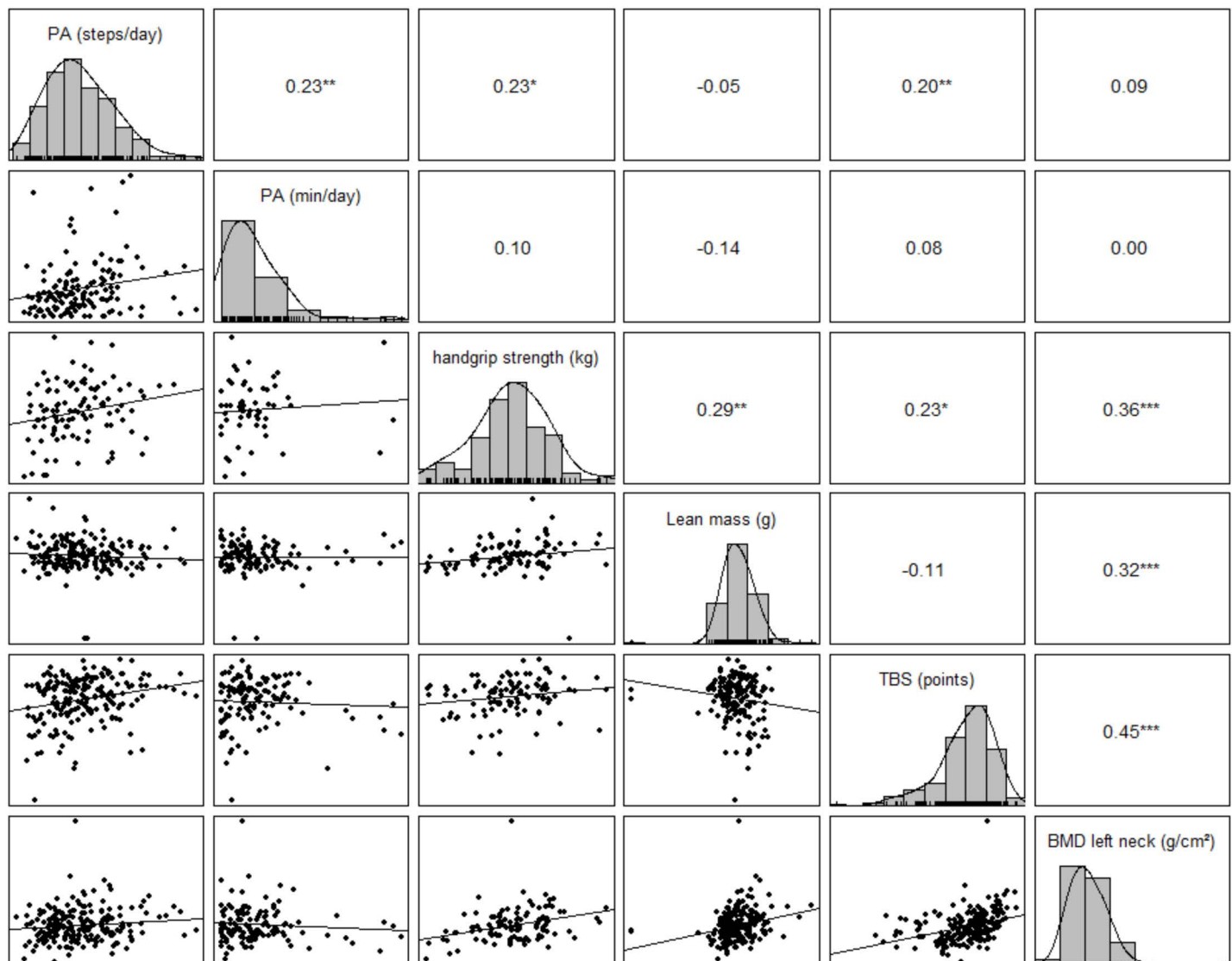

**Fig 2. Spearman's Rank correlation of physical activity on bone tissue and lean mass.** Explanation: * indicates significant difference (** p ≤ 0.01, *** p ≤ 0.001), PA (steps/day) = objective physical activity tracked via electronic activity tracking for the duration of 7 days (n = 223), PA (min/day) = self-reported subjective physical activity assessed via an activity diary for the duration of 7 days (n = 125), handgrip strength (mean value of left and right hand (n = 102)); lean mass (whole body (n = 180)); BMD = bone mineral density (n = 201), TBS = trabecular bone score (n = 194); Spearman's correlation coefficient was calculated and are displayed in the top right; individual data distribution of parameters is placed diagonally; scatter plots with regression line are shown in the bottom left part.

With respect to research question 3), a significant moderate correlation was observed between lean mass and BMD (rho = 0.309, p < 0.001), though not between lean mass and TBS (rho = -0.11, p = 0.134). The Mann-Whitney-U-Test differentiating between higher amount of lean mass and lower amount of lean mass supported these findings revealing higher BMD in patients with increased lean mass (median 0.843 [0.745; 0.978]) compared to patients with less lean mass (median 0.768 [0.670; 0.869]; p < 0.001). There was no difference between the groups with regard to the TBS (p = 0.521). Handgrip strength is positively correlated with both BMD (rho = 0.361, p < 0.001) as well as TBS (rho = 0.221, p = 0.021) and lean mass (rho = 0.287, p = 0.003). The Mann-Whitney-U-Test differentiating between higher handgrip strength and lower handgrip strength confirmed that patients with increased handgrip strength show higher BMD (median 0.841 [0.755; 0.964]) compared to patients with lower handgrip strength (median 0.772 [0.684; 0.854]; p = 0.004). The same results were seen regarding the TBS scores (lower handgrip strength: 1.427 [1.333; 1.478] versus higher handgrip strength: 1.463 [1.404; 1.503]; p = 0.020) and lean mass (lower handgrip strength: 52883 [50019; 58859] versus higher handgrip strength: 56422 [51422; 62929]; p = 0.012).

A correlation analysis was conducted to evaluate the relationship between age and subjective as well as objective activity, and handgrip strength. The analysis revealed no significant associations (p > 0.05), with the exception of a weak inverse correlation observed between age and objectively measured PA (rho = -0.164, p = 0.014).

Furthermore, a Mann-Whitney-U-Test was done to evaluate differences regarding TBS, BMD and lean mass between the patients performing strength training (n = 37) and those, who did not perform strength training (n = 88). However, no statistically significant difference was found in any of these parameters (p > 0.542).

## Discussion

Addressing three research questions, this investigation elaborated progressively on 1) the severity phenotype and its relationship to the level of PA, 2) the correlation of PA and handgrip strength with TBS, BMD, and lean mass, and 3) the impact of the lean mass and handgrip strength on BMD and TBS in PwH.

Regarding the first research question, the main finding of this investigation is that irrespective of the disease severity phenotypes the median step activity is 7392 [4981,10679] steps/day and 42 [22, 81] minutes/day, though both do not differ between severity phenotypes. It is assumed that the statistically significant difference is missed due to the high variability of the data in all three severity phenotypes. An inverse correlation between HJHS and objective PA was observed, which is in line with previous findings [40]. Though, the key finding is that PA levels in PwH are very heterogenic. This might be due to the fact that there are several influencing factors, as presence of pain, restriction of joint function or motivation [41]. However, there is only little research investigating daily step count in haemophilia, so that the present data of a large sample size (n = 223) can serve as orientation within the scientific community. Furthermore, handgrip strength was used as an indicator for overall fitness levels [36]. The present results of handgrip strength in PwH are comparable to previous findings [9,21,22]. The present study showed an inverse correlation between handgrip strength and the HJHS, though no significant difference between the severity phenotypes.

Considering the subjective activity diary results, most PwH in all severity phenotypes conducted low-impact activity such as walking or cycling. These findings are in line with previous research and are comparable to the European population [12,42].

Nonetheless, according to data of this study, 35.4% of the patients reported not being physically active at all, which is alarmingly high and elevated compared to the adult

non-haemophilic European (mean age 67.8 ± 8.9) population, where the prevalence of physical inactivity ranges from 4.9% in Sweden to 29.0% in Portugal [43]. The reason why PwH are physically inactive can be due to several reasons, including the presence of arthropathic related pain, fear of exercise-induced bleeding events or restricted movement function [6,9,44,45]. There might also be other non-haemophilia-specific barriers (i.e., time restrictions, low motivation) to be physically active, so that an individual approach by a comprehensive haemophilia care center is necessary to target the increase of PA.

Previous research further suggested a positive relationship between age and activity levels as well as handgrip strength [46]. To control for mediation effects, a correlation analysis was conducted within the present investigation, revealing that age only weakly correlates with objective PA, but not with subjective PA or handgrip strength.

To answer research question 2 and 3, associations between both objective and subjective activity levels and handgrip strength, TBS, BMD, as well as lean mass were analyzed. PA does have multiple short-term but also long-term effects such as increased lean mass, enhanced handgrip strength and a positive influence on bone remodeling [47,48]. Though, it needs to be emphasized that the recorded PA did not affect lean mass or BMD. With regard to the BMD, the results were expected as BMD needs high impact PA, e.g., in form of strength training, to adapt accordingly. However, the subanalysis of PwH performing strength training compared to PwH, who did not do strength training revealed no significant differences regarding BMD. This is most likely due to the fact that strength training can involve a broad variety of exercises and intensities, which has not been covered with the activity diary, meaning that there might be a measurement bias present within this analysis. However, it was observed that objective PA as well as handgrip strength are positively associated with TBS and handgrip strength also correlates with lean mass and BMD.

The present data show that lean mass does not affect the TBS, which agrees with previous literature [49]. Despite, the results of the present study confirmed that the higher the lean mass, the higher the BMD. This might be due to the fact that lean mass expresses osteogenic factors such as interleukin-6 and therefore positively influences bone remodeling [49–51]. Literature revealed that a combined training model involving resistance and weight-bearing training is recommended to effectively increase and individuals' lean mass and therefore positively impact the BMD [51]. Hereby, dynamic training with rather short stimulus duration and greater repetition frequency is highly suggested as this results in frequent loading and unloading through axial weight loading and muscle pulls [52]. This underscores the need of designing and monitoring an individualized training program for PwH by haemophilia care professionals. Such programs should prioritize bleeding prevention and address joint dysfunctions [7].

Strikingly, two different characteristics have made the TBS noticeable, which are highlighted in the following: First, TBS is positively associated with the recorded step activity while BMD is not. And second, TBS is rather normal in the haemophilic cohort though BMD is not [28]. The two components of bone quality (BMD and TBS) seem to vary in the magnitude of influencing factors. Impactful lifestyle factors on levels of BMD are well-known and studied, while only little research has been done on determinants of TBS. Though, research has shown that increased weight and low PA in childhood as well as the presence of diabetes or rheumatoid arthritis are associated with lower TBS in men [29,53]. The present investigation suggests that low impact PA is already sufficient to positively affect the TBS in PwH. Most of the PwH are able to conduct low impact PA and therefore promoting their TBS, which decreases the risk of fractures. Meaning, even though the patient is restricted in doing PA and shows reduced BMD, it is of major importance to promote step activity to decrease risk of fractures.

### Strength and limitations

This investigation used valid objective activity tracking, utilizing the FitBit Alta HR and objective DXA derived data, which is considered gold standard for analyzing metrics associated with body composition, i.e., lean mass, BMD, and TBS, highlighting this study's high degree of quality [54,55]. Especially the evaluation of TBS in relation to PA is one major strength as this has not been investigated previously.

However, there are noteworthy limitations to declare. The main limitation is the recording of subjective PA as the activity diary is limited in expressiveness due to low standardization. This study lacks in investigating the nature of strength training more precisely, as strength training can encompass a broad spectrum from functional training to weight-bearing exercises either whole body or only within subregions. Furthermore, within this study a 7-day evaluation of activity was generated to get an insight in patients overall activity level, though no retrospective data on activity levels are present. As bone remodeling is a time-bound process, longitudinal studies investigating PA in PwH are necessary to check for a causal relationship of PA on BMD, but also for more information on TBS in PwH [56]. The present data of BMD are derived from the hip (neck), as these data are less prone to bias compared to BMD of the spine given a high prevalence of degenerative changes of vertebrae bodies, which can lead to false positive increased BMD scores [57]. However, this does not affect the TBS, but this needs to be noted since TBS can be determined at the spine only [57]. In addition, no data on the social or occupational status was gathered for this investigation, so that no socioeconomic facilitators or barriers for PA can be evaluated.

### Conclusion

The present study showed that step activity and self-reported daily PA in minutes do not differ between the severity phenotypes. The majority of PwH in all severity phenotypes performed low-impact PA, such as walking or cycling. Most likely, these activities do not evoke sufficient strain to promote bone formation, so that no relationship between PA and BMD is observed. Handgrip strength correlates significantly with PA, lean mass and bone quality. Handgrip strength is seen as an indicator of PA level, underlining an indirect relationship between PA and the above-mentioned parameters. Moreover, there is a positive association between step activity and TBS, which is a key relevant finding as many PwH conduct walking as PA. With PwH, it is important to identify the optimal balance on PA that enhances bone quality and stimulates muscle activation, while minimizing excessive mechanical stress on the joints. Highlighting, safety measures and appropriate treatment enables PwH to conduct PA without risks of bleeding and should therefore be more promoted by respective haemophilia health care centers.

### Supporting information

**S1 Fig.** Minimal data set for boxplots showing the different severity phenotypes regarding physical activity, bone quality, lean mass and handgrip strength.
(DOCX)

### Author contributions

**Conceptualization:** Johannes Oldenburg, Frank Alexander Schildberg, Andreas Strauss.

**Data curation:** Pia Ransmann, Jamil Hmida, Georg Goldmann, Robert Ossendorff, Andreas Strauss.

**Funding acquisition:** Andreas Strauss.

**Investigation:** Pia Ransmann, Marius Brühl, Jamil Hmida, Johannes Oldenburg, Robert Ossendorff, Andreas Strauss.

**Methodology:** Pia Ransmann, Marius Brühl, Jamil Hmida, Georg Goldmann, Johannes Oldenburg, Robert Ossendorff, Andreas Strauss.

**Project administration:** Thomas Hilberg, Andreas Strauss.

**Resources:** Andreas Strauss.

**Software:** Alexander Schmidt, Andreas Strauss.

**Supervision:** Georg Goldmann, Johannes Oldenburg, Frank Alexander Schildberg, Fabian Tomschi, Thomas Hilberg, Andreas Strauss.

**Validation:** Robert Ossendorff, Fabian Tomschi, Alexander Schmidt, Thomas Hilberg.

**Visualization:** Alexander Schmidt.

**Writing – original draft:** Pia Ransmann.

**Writing – review & editing:** Marius Brühl, Jamil Hmida, Frank Alexander Schildberg, Thomas Hilberg, Andreas Strauss.

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
