## [Decision Letter · Decision Letter 0]

5 Jan 2025

PONE-D-24-55115Physical activity and handgrip strength in patients with mild, moderate and severe haemophilia: Impacts on bone quality and lean massPLOS ONE

Dear Dr. Strauss,

Thank you for submitting your manuscript to PLOS ONE. After careful consideration, we feel that it has merit but does not fully meet PLOS ONE’s publication criteria as it currently stands. Therefore, we invite you to submit a revised version of the manuscript that addresses the points raised during the review process.

We look forward to receiving your revised manuscript.

Kind regards,

Wolfgang Miesbach, MD

Academic Editor

PLOS ONE

Journal Requirements:

2. Thank you for stating the following financial disclosure: Bayer Vital GmbH  

3. Thank you for stating the following in the Acknowledgments Section of your manuscript: This study received financial support by Bayer Vital GmbH.

Please remove any funding-related text from the manuscript and let us know how you would like to update your Funding Statement. Currently, your Funding Statement reads as follows: Bayer Vital GmbH

Reviewers' comments:

Reviewer's Responses to Questions

**Comments to the Author**

1. Is the manuscript technically sound, and do the data support the conclusions?

Reviewer #1: Partly

Reviewer #2: Yes

2. Has the statistical analysis been performed appropriately and rigorously? 

Reviewer #1: No

Reviewer #2: Yes

3. Have the authors made all data underlying the findings in their manuscript fully available?

Reviewer #1: Yes

Reviewer #2: Yes

4. Is the manuscript presented in an intelligible fashion and written in standard English?

Reviewer #1: Yes

Reviewer #2: Yes

5. Review Comments to the Author

Reviewer #1: In this study patients with hemophilia A or B and various severity types were investigated with regard to their sports activities and association with bone mineral density (BMD) and trabecular bone score (TBS). Most interesting that differences between the severities were not seen.

I have several comments:

The authors report on 223 included patients, but less than half of them had a handgrip strength test and not all returned the diary. Of those who returned the diary, some reported that they did not perform any sports.

A table that lists all the investigations with numbers of patients in whom the parameters were collected should be given. How did the authors handle missing values in the statistical evaluation?

Provide numbers of hemophilia A and B in table 1.

The joint score HJHS differed – as expected – between hemophilia severities, but was not fully included in the analyses. Was there any associations of outcomes (BMD and TBS) with HJHS?

Table 2: It would be fine to also include percentage values.

Page 8. Line 4: …post-hoc testing of joint score. Why post-hoc testing? Please explain.

Page 17, first line: “…still being gentle the joints” Please clarify or correct

Reviewer #2: Haemophilia A and B can be associated with limited mobility and physical fitness due to degenerative joint and bone problems. Further investigation of this relationship would be useful for creating a coordinated individual treatment plan.

Some points could be clarified:

Since only the HJHS significantly differed between the different degrees of severity, it would be interesting to know if this had an influence.

As there is no healthy control group, what would the parameters look like in a population without haemophilia?

Not all patients followed the sports activities. What were the reasons for this?

Could some information on social status and occupation be added?

There is a wide age range. Is the younger population different from the older one?

6. PLOS authors have the option to publish the peer review history of their article (what does this mean? ). If published, this will include your full peer review and any attached files.

**Do you want your identity to be public for this peer review?** For information about this choice, including consent withdrawal, please see our Privacy Policy .

Reviewer #1: No

Reviewer #2: No

---

## [Author Response · Author response to Decision Letter 1]

14 Jan 2025

Reviewer #1:

1. The authors report on 223 included patients, but less than half of them had a handgrip strength test and not all returned the diary. Of those who returned the diary, some reported that they did not perform any sports.

A table that lists all the investigations with numbers of patients in whom the parameters were collected should be given. How did the authors handle missing values in the statistical evaluation?

Thank you for this valuable input. It is necessary that the sample sizes are clearly determined. We now included the number of patients for each parameter in the legends of the figures (see figure 1 & 2). As this is a cohort study with multiple outcome parameters, it is challenging to completely avoid missing data. For the statistical analysis, we addressed this by excluding the missing data to ensure the robustness of our findings.

2. Provide numbers of hemophilia A and B in table 1.

Thank you very much for this advice. We included the respective numbers in table 1.

3. The joint score HJHS differed – as expected – between hemophilia severities but was not fully included in the analyses. Was there any associations of outcomes (BMD and TBS) with HJHS?

Thank you for this question. The data of the present manuscript were part of a large investigation. We previously published data regarding the HJHS and BMD, indicating that HJHS predicts BMD (B= −0.002; 95% CI: −0.004, −0.001;

p = 0.02). To avoid duplication of data presentation, these results were not mentioned in the present manuscript again.

4. Table 2: It would be fine to also include percentage values.

Thank you for your valuable feedback. The percentage values are now included in table 2.

5. Page 8. Line 4: …post-hoc testing of joint score. Why post-hoc testing? Please explain.

Thank you for your comment. To be more precise, we added “Bonferroni” post-hoc-testing in the text (see page 8, line 160) As the Kruskal-Wallis-Test showed a significant difference of p=0.001 for the severity phenotype regarding joint score, a post-hoc testing was used for (1) the calculation pairwise comparisons and (2) to adjust the p-value to minimize the type-I error rate.

6. Page 17, first line: “…still being gentle the joints” Please clarify or correct.

Thank you for your valuable feedback. To be clearer, we adjusted the sentence to: “while minimizing excessive mechanical stress on the joints” (see page 18, line 419).

Reviewer #2:

1. Since only the HJHS significantly differed between the different degrees of severity, it would be interesting to know if this had an influence.

Thank you very much for this comment. As stated on page 13, line 322 as well as page 14, line 330, the HJHS was significantly correlated with objective PA and handgrip strength. Indicating that the worse the joint status, the lower the handgrip strength and the lower the objective PA. For subjective PA, no significant correlations were found. Furthermore, the data of the present manuscript were part of a large investigation. We previously published data regarding the HJHS and BMD, indicating that HJHS predicts BMD (B= −0.002; 95% CI: −0.004, −0.001; p = .02). To avoid duplication of data presentation, these results were not mentioned in the present manuscript again (see doi: 10.1016/j.rpth.2024.102624 & doi: 10.1111/hae.15091).

2. As there is no healthy control group, what would the parameters look like in a population without haemophilia?

Thank you for your question. Depending on the outcome parameter, we included comparisons to the non-haemophilic European population (PA)

(see page 14, line 334, and 337). Regarding bone quality and body composition parameters, the comparisons to a healthy population were already stated in previously published papers, so that we did not mention these data again to avoid duplication (see doi: 10.1016/j.rpth.2024.102624 & doi: 10.1111/hae.15091).

3. Not all patients followed the sports activities. What were the reasons for this?

Thank you for highlighting this point. Some haemophilia-specific reasons might be presence of pain, restriction of joint function or fear of exercise-induced bleeding events (as stated on page 14, line 338-341). However, an evaluation of possible barriers to perform sports was not conducted within this research. Of course, there might also be non-haemophilia specific barriers to perform sports. We added this paragraph in the discussion section including a respective reference (doi: 10.3390/ijerph18115810; see page 14, line 341-343).

4. Could some information on social status and occupation be added?

This is a very good idea, thank you. Unfortunately, within this research project we did not gather data on social status/occupation of the patient. We included this aspect in the limitations section (see page 17, line 405-407).

5. There is a wide age range. Is the younger population different from the older one?

Thank you for this valuable question. Based on an additional correlation analysis we could demonstrate that age does not impact handgrip strength nor subjective PA. Objective PA was weakly correlated with age (rho= -0.164, p=0.014). We added this finding in the results section (page 13, line 303-306) and discussed this in on page 14, line 344-347 including a respective reference (doi: 0.1186/s12877-021-02188-9).

---

## [Decision Letter · Decision Letter 1]

11 Feb 2025

Physical activity and handgrip strength in patients with mild, moderate and severe haemophilia: Impacts on bone quality and lean mass

PONE-D-24-55115R1

Dear Dr. Strauss,

We’re pleased to inform you that your manuscript has been judged scientifically suitable for publication and will be formally accepted for publication once it meets all outstanding technical requirements.

Kind regards,

Wolfgang Miesbach, MD

Academic Editor

PLOS ONE

Reviewers' comments:

Reviewer's Responses to Questions

**Comments to the Author**

1. If the authors have adequately addressed your comments raised in a previous round of review and you feel that this manuscript is now acceptable for publication, you may indicate that here to bypass the “Comments to the Author” section, enter your conflict of interest statement in the “Confidential to Editor” section, and submit your "Accept" recommendation.

Reviewer #2: All comments have been addressed

2. Is the manuscript technically sound, and do the data support the conclusions?

Reviewer #2: Yes

3. Has the statistical analysis been performed appropriately and rigorously? 

Reviewer #2: Yes

4. Have the authors made all data underlying the findings in their manuscript fully available?

Reviewer #2: Yes

5. Is the manuscript presented in an intelligible fashion and written in standard English?

Reviewer #2: Yes

6. Review Comments to the Author

Reviewer #2: (No Response)

7. PLOS authors have the option to publish the peer review history of their article (what does this mean? ). If published, this will include your full peer review and any attached files.

**Do you want your identity to be public for this peer review?** For information about this choice, including consent withdrawal, please see our Privacy Policy .

Reviewer #2: No

---

## [Editor Report · Acceptance letter]

PONE-D-24-55115R1

PLOS ONE

Dear Dr. Strauss,

I'm pleased to inform you that your manuscript has been deemed suitable for publication in PLOS ONE. Congratulations! Your manuscript is now being handed over to our production team.

Kind regards,

on behalf of

Dr. Wolfgang Miesbach

Academic Editor

PLOS ONE